# Assessment of the Seismic Behavior of Selective Storage Racks Subjected to Chilean Earthquakes

**Eduardo Nuñez [1],*, Catalina Aguayo [1] and Ricardo Herrera [2]**

[1] Department of Civil Engineering, Universidad Católica de la Santísima Concepción, Concepción 78349, Chile; caguayoo@magister.ucsc.cl

[2] Department of Civil Engineering, Universidad de Chile, Santiago de Chile 8370449, Chile; riherrer@ing.uchile.cl

* Correspondence: enunez@ucsc.cl; Tel.: +56-9-51277382

**Abstract:** A seismic performance evaluation of selective storage racks subjected to Chilean Earthquakes was conducted using nonlinear pushover and nonlinear dynamic time-history analyses. Nine seismic records with two horizontal components and magnitude Mw > 7.7 were applied to numerical models of prototype rack structures. The prototype racks were designed considering two types of soil and two aspect ratios. The inelastic behavior of beam connections was included in the models. The results showed a predominantly elastic behavior, mainly in the cross-aisle direction, in comparison to the down-aisle direction. The inelastic action was concentrated in pallet beams and up-rigths. Higher values of base shear were reached, due to elevated rigidity in rack configurations, and an acceptable performance was obtained. A response reduction factor was reported in both directions, reaching values larger than the limit imposed by the Chilean standard. However, values below this limit were obtained in the cross-aisle direction, in some cases. Finally, in all cases, the calculated response modification factor is highly influenced by the overstrength obtained from seismic design.

**Keywords:** storage racks; seismic performance; cold formed steel; brace; moment connection

## 1. Introduction

Steel storage rack systems are commonly used to store different types of loads. The rack structures are loaded for long periods of time, therefore, the likelihood that they are loaded during a seismic event is high. In Chile, selective storage systems are preferred by owners over other types of racks. Aditionally, Chile is one of the most seismically active countries in the world, having sustained several large earthquakes that produced significant human and material losses. However, there is limited information associated with the seismic performance of rack systems during Chilean earthquakes. Furthermore, the low buckling resistance of thin sections normally used in this type of structure results in low strength and ductility in cold-formed structural elements, which can limit their performance under seismic actions. Semi-rigid beam-end connections have been used in the structural design, providing large local and global deformations, which require consideration without the exception of second order effects. Consequently, the use of slender rack configurations in soft soils could result in poor seismic performance affecting the parameters of seismic design, such as the response reduction factor and ductility. A brief description of previous research related to the seismic performance of industrial steel storage racks is presented next.

Kanyilmaz et al. [1] studied the behavior of braced steel storage rack systems, performing full-scale pushover testing specimens representative of these systems. This study concluded that, during design, sufficient overstrength should be guaranteed for the bracing connections, in order to avoid a global brittle collapse. A numerical study of cold-formed steel storage racks with spine bracings using

five types of speed-lock connections with bolts was conducted by Yin et al. [2]. Single entry and double entry rack configurations were analyzed, incorporating plastic hinges in pallet beams with properties obtained from experimental tests. Pushover analyses were performed on the rack models, to investigate their seismic response. The results highlighted that the upper bolt and welds were more effective in improving the seismic performance compared to the lower bolt in end connection beam. Later, Yin et al. [3] evaluated the seismic performance of cold-formed steel storage racks with spine bracings, based on the nonlinear dynamic response history analysis as an extension of their previous research [2], considering ground motion records from the PEER database. The analyses included incremental dynamic analysis (IDA). The collapse mechanisms were studied from the IDA results and compared to the pushover analysis results.

Kanyilmaz et al. [4] conducted an experimental assessment of the seismic behavior of unbraced steel storage pallet racks. The full-scale pushover test was performed in the down-aisle direction on fully-loaded unbraced specimens. The study provided design guidelines to guarantee a global ductility under seismic actions, with requirements for the design of unbraced rack connections. An advanced design procedure for the safe use of steel storage pallet racks in seismic zones was performed by Bernuzzi et al. [5]. In this study, a non-linear time history (NLTH) analysis considering low-cycle fatigue (LCF) with damage distribution was implemented. A practical case of study related to a medium-rise double-entry pallet rack was worked out, considering two Italian earthquakes and two models representing the cyclic response of beam-to-column joints. Mojtabaei et al. [6] conducted an analytical and experimental study on the seismic performance of cold-formed steel (CFS) moment frames, by testing a half-scale CFS moment-resisting portal frame under static monotonic loading up to failure. Damage was mostly concentrated at the top and bottom of the CFS bolted connection zones. The proposed system can provide adequate seismic performance, if an appropriate design of the main structural elements is achieved. The energy dissipation capacity and the ductility ratio of the proposed system increased significantly by decreasing the width-to-thickness ratio of the columns.

Maguire et al. [7] studied the cross-aisle seismic performance of selective storage racks. The uplifting and rocking behavior of three full-scale selective racks was examined considering three baseplate types: ductile, heavy-duty, and unanchored. At 1.5 times the respective design level ground motions, the heavy-duty baseplates caused a foundation failure, while the unanchored rack failed by overturning. The rack with ductile baseplates survived the test up to 2.3 times the design level. For a given ground motion, the unanchored rack upright always had the smallest peak axial load. Bernuzzi et al. [8] performed a study focused on the development of more reliable approaches for designing racks against earthquakes, with a comparison of advanced vs. standard design strategies. A wide range of cases for routine design were defined, which comprised racks differing in terms of geometric layout and component performance. For each of them, the load carrying capacity corresponding to different values of the peak ground acceleration was evaluated using two alternative design approaches: the modal response spectrum analysis (MRSA) and an advanced strategy combining non-linear time-history analyses.

Jacobsen et al. [9] conducted an experimental program consisting of quasi-static cyclic, pull-back and seismic shake table test to examine the inelastic seismic response of cold-formed selective rack structures. In the seismic tests, racks could sustain up to 10% drifts without collapse. A numerical model was proposed to predict the rack seismic response, including pallet sliding. It was used to study the response of six-bay racks with three to six levels. Castiglioni et al. [10] conducted a dynamic shake-table test to study the sliding behavior of pallets in industrial rack systems under earthquake-induced actions. Dynamic tests were performed using three beam types with different surface finish materials, in the cross-aisle (CA) and down-aisle (DA) directions. Several phenomena related to deformations of the supporting beams were observed, which affected the pallet behavior in both the CA and DA directions, with sliding at very low accelerations levels. The same response was observed for uniaxial earthquake tests. For biaxial seismic testing, lower bound acceleration in the CA direction was higher than in dynamic cyclic tests, whereas the opposite was observed in the DA direction. Gusella et al. [11] evaluated the influence on the structural response of rack connections due to the

structural details, randomness in the geometrical features, and mechanical properties of connection members (beam, weld, connector, and column). A Monte Carlo simulation was conducted, to explore the impact of variability in design parameters on the initial flexural stiffness and ultimate flexural capacity of rack connections. Variability in member geometrical features was determined from current design specifications, while variability in steel mechanical properties was determined by experimental tests. Results indicated that system effects reduce flexural stiffness, and the variability in the response of individual components does not propagate to the overall flexural capacity.

Dubina et al. [12] studied the modelling of impact for progressive collapse assessment of selective rack systems (SPR). Structural configurations were varied to consider different connection properties and bracings. SPR structures are susceptible to global failure, especially if the spine bracing is in just few spans, and the connection rigidity is low. The explicit modelling of the forklift impact provided the most accurate results. Non-linear static pushdown analysis can provide satisfactory results at the least computational effort, whereas dynamic increase may require corrections. Jovanović et al. [13] studied a hysteresis model for beam-to-column connections of steel storage racks, developed with a new constitutive model. By using only four parameters, the model can simulate both pinching and low-cycle fatigue of a beam-to-column connections (BCC). The model predictions were evaluated through a comparison of dissipated energy and resisting moment through cycles, and were proven satisfactory. The non-linear time history analyses were performed on three identical racks with different BCC constitutive models, including the proposed one. These analyses validated the proposed model as computationally effective under dynamic loading. Aguirre [14] conducted an experimental study of the seismic behavior of rack structures, tested under static and cyclic loads. The beam-to-column connection studied utilized hooks fabricated with the beam, inserted in special slots at the columns. The moment-rotation curves obtained showed a continuous hardening, enabling the connections to reach about half of the beam plastic moment. The failure mode was initiated with yielding of the outermost hooks in a progressive sequence towards the beam neutral axis. Non-linear analyses of the rack structure under different seismic conditions were performed, including one record of a Chilean earthquake.

In summary, limited research of seismic performance of selective racks subjected to Chilean earthquakes have been performed. A more systematic and thorough evaluation of the seismic performance of these systems is required, in order to prevent the failures observed in previous events from occuring again, and to establish a rational set of seismic design parameters for selective racks. This study aims to investigate the seismic performance of steel storage selective racks subjected to Chilean earthquakes, using a state of the art methodology for steel structural systems. Rack structures with different heights and spans in the down-aisle direction are analyzed and two soil types are considered. These soil types are classified as soft soils according to NCh2369 [15]. First, a seismic design of rack structures was performed according to [15], and then the inelastic response was studied using nonlinear pushover and time-history analyses. Several earthquake records were used, and the nonlinear behavior of the connections obtained from experimental tests reported in the literature was considered. Finally, seismic parameters such as the reduction factor R, overstrength, and ductility were obtained and compared to the design values available.

## 2. Description of Rack Structures Considered

Steel storage racks are used to store goods in warehouses. There are different types of racks, out of which the selective racks systems have been widely used in Chile. Their main advantage is the use of a first-in first-out (FIFO) system, allowing a fast movement of goods. The structural system of selective racks is composed by vertical members (up-rights) and horizontal members (pallet beams). The up-rigths are made of perforated thin walled sections, the pallet beams are made from built-up closed sections and bracings are used in cross-aisle direction, as shown in Figure 1. Their elements are designed according to a specification for cold-formed members [16]. Generally, the seismic force resisting system is different for the two resisting directions: i) braced frame in the cross-aisle direction; and ii) moment frame in the down-aisle direction. The beam-column joint behavior plays a fundamental role in the seismic performance of rack structures. The behavior of moment connections

has been extensively studied. Pinching and limited capacity of energy dissipation is characteristic in these types of connections.

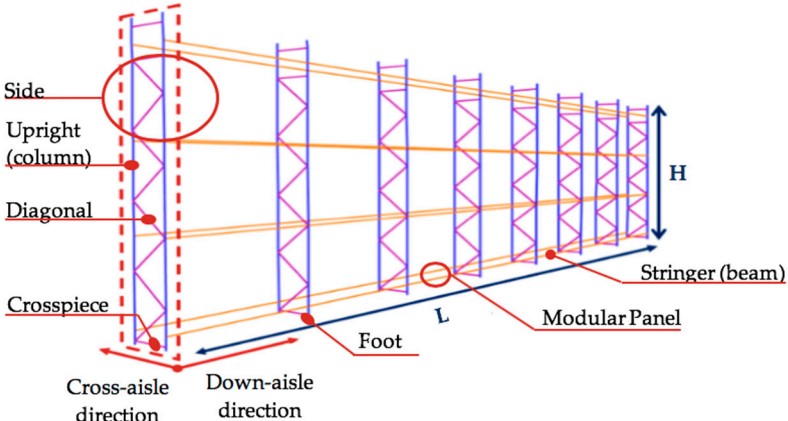

**Figure 1.** Steel storage selective rack schematic diagram.

A relevant parameter in the seismic design of rack structures is the global slenderness or aspect ratio in each direction. Slender racks are susceptible to overtuning or excessive frame distortion, decreasing their efficiency against seismic loads. In this study, a seismic assessment of selective rack systems with different global slendernesses was performed. The slenderness parameter was considered separately for both directions, i.e., a cross-aisle slenderness ($\lambda_{CA}$) and a down-aisle slenderness ($\lambda_{DA}$). A summary of the slenderness parameters by model of rack structure studied is shown in Table 1. In Figure 2, views of the five types of selective rack structures studied are shown: SelCB is a model of selective rack with a short-low slenderness relationship, where S3 or S4 indicates the soil type, according to [15]; SelCA is a model of selective rack with short-high slenderness relationship; SelLB is a model of selective rack with large-low slenderness relationship; and SelLA is a model of selective rack with large-high slenderness relationship. These models were designed considering two soft soils, as discussed in Section 3.

**Table 1.** Summary of slenderness parameters in rack structures studied.

| Model | H (m) | L (m) | B (m) | $\lambda_{CA}$ | $\lambda_{DA}$ |
|-------|-------|-------|-------|------|------|
| SelCB | 6.51 | 18.90 | 0.75 | 8.7 | 0.34 |
| SelCA | 12.21 | 18.90 | 0.75 | 16.4 | 0.65 |
| SelLB | 6.51 | 35.10 | 0.75 | 8.7 | 0.19 |
| SelLA | 12.21 | 35.10 | 0.75 | 16.4 | 0.34 |

Regarding the material properties of each member, the nominal yield stress and strength were employed considering steel grade ASTM A36 [17]. The A36 material is commonly used in the fabrication of racks manufactured in Santago de Chile, Chile. These values are indicated in Table 2.

**Table 2.** Material properties.

| Element | Fy (MPa) | Fu (MPa) | E (MPa) |
|---------|----------|----------|---------|
| Up-rights, pallet beams, and brace members | 253 | 408 | 200,000 |

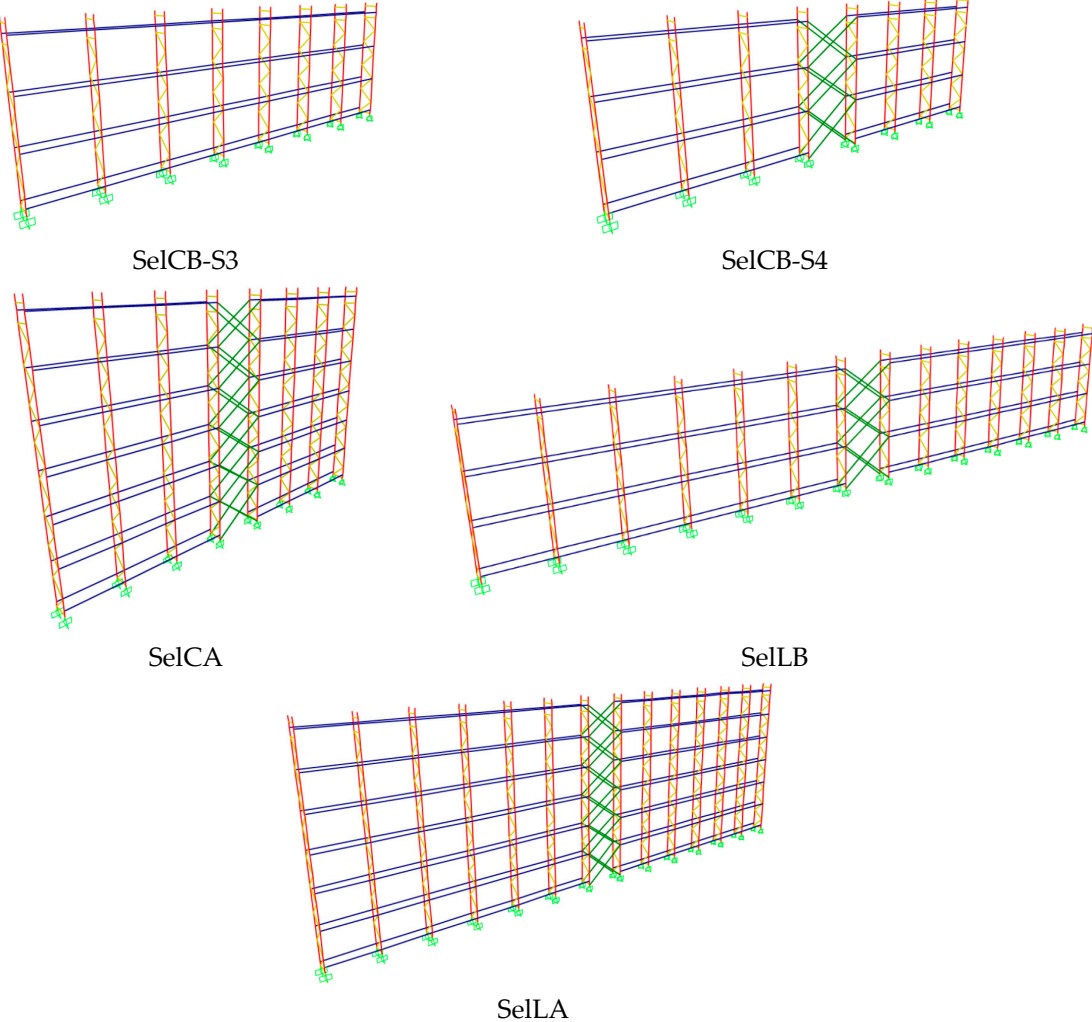

SelCB-S3     SelCB-S4

SelCA     SelLB

SelLA

**Figure 2.** View of 3D rack models.

## 3. Design of Rack Structures Considered

A seismic design with modal response spectrum analysis (MSRA) of the propotype models was performed according to [15]. Loads, load combinations, story drift limits, and design spectrum were obtained from this standard. The characteristics of models, loads, and results are described here.

### 3.1. Numerical model

The rack structures were modeled and analyzed using the software SAP2000 v22 [18]. The members were modeled using frame elements with two end nodes and six degrees of freedom per node. The base connections were idealized as fully restrained base columns in the down-aisle direction, considering the elastic stiffness, according to experimental tests [19]. In the cross-aisle direction a pinned base was used. The moment connections were considered according to [20], and five types of moment connections were considered, out of which only the JD5 (Speed-lock connection with two bolts and welded all around, according to [20]) moment connection achieved a good performance, unlike the other connections. Therefore, this connection was selected for the study. The elastic stiffness of the JD5 moment connection was incorporated in the numerical models, using the results of the parametric study performed in [20]. Finally, the bending stiffness in braces was released at both start and end of the element. Later, a design of members was performed according to [16], using an algorithm developed by the authors. The second-order effects were considered with P-delta plus large displacements analysis, introducing a previous nonlinear case. All load cases were analyzed from this P-delta case. Bracing towers were required in some models, to comply with the story drift limits established in [15].

### 3.2. Loads

Permanent and live loads were considered. A uniformly distribuited gravity load of 3.6 kN/m was applied to the pallet beam. In summary, two units loads of storage racks were used in pallet beams (unit load = 9.8067 kN). The mass source was considered as 100% dead plus 50% units loads. However, a different reduction factor of unit loads may be considered, according to [10].

Rgarding seimic loads, one seismic zone (Ao = 0.4g) and two soft soils (soil types 3 and 4, according to [15]) were analyzed. The design spectrum with R = 1 for each soil condition is shown on Figure 3. On other hand, a response reduction factor R = 4 is specified in seismic code [15] for steel storage racks.

### 3.3. Results of Seismic Design

A list of geometric properties and section shapes obtained from the seismic design are shown in Table 3. For the SelCBS3 model, a braced tower was not required; however, for the other models, it was neccesary to comply with the drift limit requirement established in [15].

**Table 3.** Geometric properties of rack members (the dimensions are indicated in millimeters).

| Element Type | Cold-Formed Section | | | | | | | |
|---|---|---|---|---|---|---|---|---|
| | **SelCBS3** | **SelCBS4** | **SelCAS3** | **SelCAS4** | **SelLBS3** | **SelLBS4** | **SelLAS3** | **SelLAS4** |
| Up-rights | TX 100 × 105 × 3 | TX 100 × 105 × 3 | TX 160 × 105 × 3 | Double TX 100 × 105 × 3 | TX 100 × 105 × 3 | TX 100 × 105 × 3 | Double TX 140 × 105 × 3 × 2, 5 | Double TX 140 × 105 × 3 × 2, 5 |
| Pallet beam | TC 125 × 50 × 2 | TC 125 × 50 × 2 | TC 125 × 50 × 2 | TC 125 × 50 × 2 | TC 125 × 50 × 2 | TC 125 × 50 × 2 | TC 125 × 50 × 2 | TC 125 × 50 × 2 |
| Brace | C 58 × 25 × 2 | C 58 × 25 × 2 | C 58 × 25 × 2 | CA 70 × 26 × 10 × 2 | CA 70 × 26 × 10 × 2 | CA 70 × 26 × 10 × 2 | CA 70 × 26 × 10 × 2 | CA 70 × 26 × 10 × 2 |
| Braced Tower | - | Tube 75 × 75 × 2 | Tube 75 × 75 × 2 | Tube 75 × 75 × 2 | Tube 75 × 75 × 2 | Tube 75 × 75 × 2 | Tube 75 × 75 × 2 | Tube 75 × 75 × 2 |

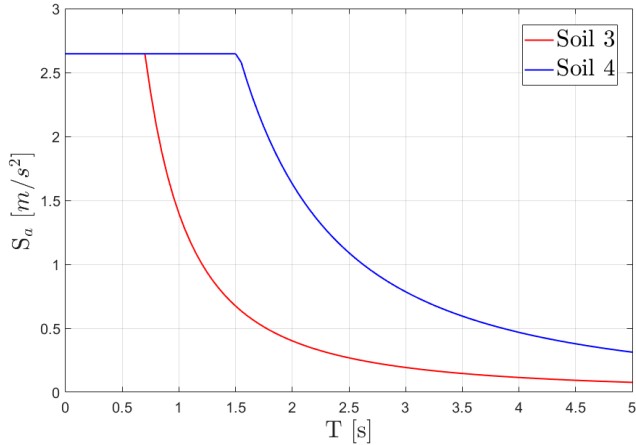

**Figure 3.** Design spectrum according to NCh2369 [15].

In order to characterize the global behavior in the down-aisle and cross-aisle directions, fundamental periods, seismic weight and base shear are shown in Table 4. The braced tower reduces the period values. However, it impacts in the base shears which were increased in the down-aisle direction. On other hand, the design base shears are approximately 20%, in comparison to the seismic weight design in the models, except in the SelCBS3 model, where the seismic behavior is controlled by the moment frame in down-aisle direction.

**Table 4.** Period, Seismic Weight and Base Shear in rack models.

| Model | $Tx$ [s] | $Ty$ [s] | $P$ [kN] | $Q_{min}$ [kN] | $Q_{max}$ [kN] | $Q_{dx}$ [kN] | $Q_{dy}$ [kN] |
|---|---|---|---|---|---|---|---|
| SelCBS3 | 1.66 | 0.33 | 265.90 | 26.59 | 71.79 | 26.59 | 40.72 |
| SelCBS4 | 0.18 | 0.33 | 230.72 | 23.07 | 62.30 | 42.21 | 30.44 |
| SelCAS3 | 0.45 | 0.89 | 407.64 | 40.76 | 110.06 | 72.99 | 41.56 |
| SelCAS4 | 0.38 | 0.70 | 419.30 | 41.93 | 113.21 | 77.24 | 55.39 |
| SelLB | 0.25 | 0.32 | 458.30 | 45.83 | 123.74 | 84.33 | 68.70 |
| SelLA | 0.50 | 0.69 | 834.42 | 83.44 | 225.29 | 151.20 | 124.91 |

## 4. Nonlinear analysis

The nonlinear analysis has been used in structural engineering as a tool to perform the seismic assessment of buildings and industrial structures. The principal advantage of the nonlinear static pushover analysis is that it is simple to implement, obtaining a structural response at different levels of demand, achieving a solution in which the structure is permanently in static equilibrium. However, the disadvantages of the pushover method are that the solution for target displacement is approximate, the results of higher mode effects or cyclic degradation are not represented, and the cyclic response from ground motions is lost.

On the other hand, the nonlinear dynamic analysis is a more refined method than nonlinear static analysis. Furthermore, ground motion records and cyclic behavior of elements are required to implement it. Higher mode effects, as well as hysterical degradation, are considered in nonlinear dynamic analysis, representing the main advantage of this analysis compared to static nonlinear analysis. However, the structural response obtained is directly related with the hysteretic elements and ground motions applied, which introduces uncertainties. Therefore, it is necessary to use a large number of records, and to obtain demands considering the statistical variation derived from these records.

In rack structures, a highly nonlinear response is expected. However, the seismic design is performed using the typical modal response spectrum analysis in the elastic range; therefore, the inelastic behavior is not obtained. As follows, the inelastic response of rack structures are performed using the nonlinear pushover analysis and nonlinear dynamic analysis.

### 4.1. Numerical Model

The nonlinear response was evaluated using the numerical models previously utilized in seismic design, incorporating plastic hinges in members. The plastic hinges were defined according to concentrated damage, based on ATC-40 [21] and FEMA356 [22] (see Figure 4a). In this study, different hinge models were employed according to members. Fiber-based hinges considering P-M2-M3 interaction (axial-primary momento-secondary moment) were used in the up-rights and the elastic stiffness was introduced [2]. A fiber-based hinge with axial only behavior was used in the brace elements, in accordance with [2]. The plastic hinges in pallet beams are commonly modeled ([23–25]) using an M-hinge model without axial force, similar to FEMA 356 model [22]. In this research, the moment-rotation (M-θ) relationship was parametrized from experimental studies in [20]. According to [14], the backbone curves of JD5 moment connection reached the best performance due to welding around the beam and the use of three bolts, unlike the other connections. Furthermore, a pivot hysteretic model proposed by Dowel et al [26] was used to capture the cyclic response of moment connections (see Figure 4b). Similarly, the pivot hysteresis model was used by [3]. The parameters of the pivot model obtained by Yin et al. [3] are summarized as follows: $\alpha1$ = 100; $\alpha2$ = 100; $\beta1$ = 0; $\beta2$ = 1; $\eta$ = 0. The parameters of the normalized backbone curve are presented in Table 5. The hysteretic behavior was used in nonlinear time history analysis. P-delta effects were considered in nonlinear pushover and nonlinear time history analysis. In the Figure 5a, the hysteretic response of the moment connection in experimental study according to [3] is shown. In the Figure 5b, the hysteretic response of moment connection with TC125 × 50 × 2 beam used in SAP2000 is shown. Additionally, the parameters of the pivot model used in SAP2000 are summarized as follows: $\alpha1$=100; $\alpha2$ = 100; $\beta1$ = 0; $\beta2$ = 0.12; $\eta$ = 0. A higher elastic stiffness was obtained in the numerical model due to greater size of beam used in comparison to experimental study (TC100 × 50 × 1.5); however, the hysteretic behavior was captured.

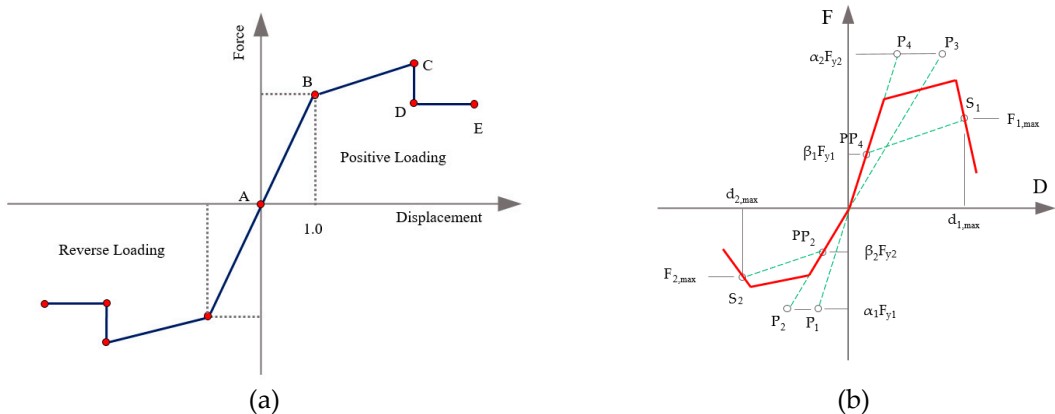

(a)      (b)

**Figure 4.** (**a**) Typical force-deformation curve according to FEMA 356 [22] and (**b**) parameter definition of the pivot hysteresis model adapted from [3], with permission from Elsevier, 2020.

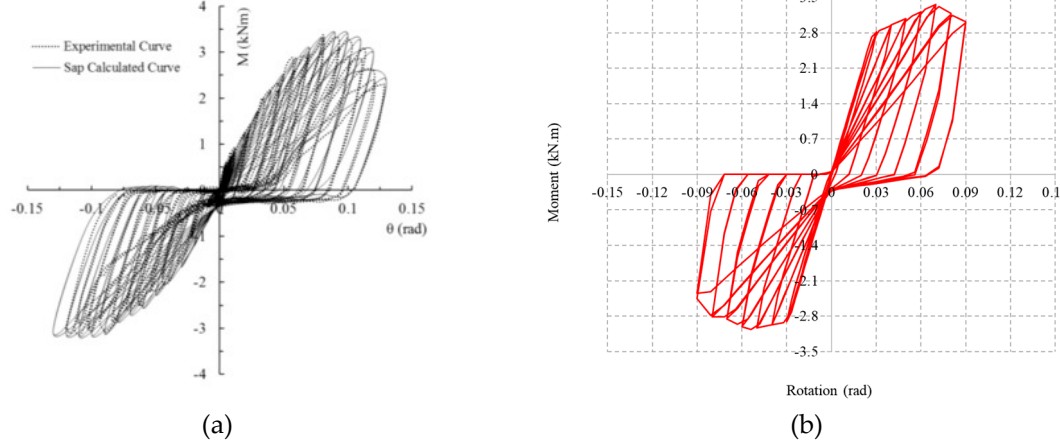

(a)      (b)

**Figure 5.** (**a**) Hysteretic response of moment connection JD5, according to experimental study reproduced from [3] (with permission from Elsevier, 2020) with type of beam TC100 × 50 × 1.5 (mm), and (**b**) hysteretic response of moment connection considered in SAP2000 using a pivot model with type of beam TC125 × 50 × 2 (mm).

**Table 5.** Parameters of the characteristic points from the backbone curve (YIN).

| Load Direction | Test Value | | Normalization | A | B | C | D | E |
|---|---|---|---|---|---|---|---|---|
| Positiva | θy/rad | 0.0490 | θ/θy | 0 | 1 | 1.8160 | 2.2780 | 2.4820 |
| | My/kN.m | 2.7896 | M/My | 0 | 1 | 1.2040 | 1.0840 | 1.0840 |
| Negativa | θy/rad | 0.0769 | θ/θy | 0 | 1 | 1.3460 | 1.6170 | 1.7470 |
| | My/kN.m | 2.9057 | M/My | 0 | 1 | 1.0560 | 0.9670 | 0.9670 |

*4.2. Nonlinear Static Pushover Anlyses*

The first approach to the evaluation of the seismic performance of the selective rack structures was performed using nonlinear static pushover analyses. A triangular load pattern equivalent to the seismic force was applied in the down-aisle and cross-aisle directions, according to FEMA 440 [27]. Furthermore, a uniformly distributed gravity of 3.6 kN/m was applied to pallet beams. On the other hand, a node control on the top of rack structures was set as the control node for displacement verifications. The target displacement was based on coefficients method, according to FEMA 356 [22]. Consequently, the capacity curves were defined as base shear vs. displacement of single degree-of-freedom (SDOF) oscillator, equivalent to the 3D structure. The acceleration-displacement response spectrum method (ADRS) was used, in accordance with ATC-40 [21]. The performance point was obtained from the intersection between capacity curve and spectrum demand. The seismic design parameters, such as the response reduction factor $R = R_\mu R_\Omega$ were determined, where $R_\mu = V_e/V_{max}$, is the ductility reduction factor (elastic shear divided by maximum shear), and $R_\Omega = V_{max}/V_d$, is the overstrength of the structure (maximum shear divided by design shear). Finally, the ductility factor $\mu = \Delta_{max}/\Delta_y$ was obtained, dividing the maximum displacement by the yield displacement.

*4.3. Results of Pushover Analysis*

The capacity curves from pushover analysis are plotted in Figure 6. The rack models reached an elastic behavior in cross-aisle direction, while in the down-aisle direction, an inelastic behavior was obtained. Large displacements were reached by the SelCBS3 model, in comparison to the other models, due to the flexibility of the moment frame, unlike the models with bracing towers. Aditionally, the elastic slope of SelCBS3 curve is smaller in comparison to SelCBS4, which exhibits high elastic stiffness. Moreover, the SelLB model reached a higher stiffness in comparison to the SelLA model, showing the incidence of slenderness in global response of racks. Comparing the response of rack models, those designed for S3 soil exhibited lower stiffness and displacement than models designed for S4 soil, demonstrating the influence of soil in the rack design. Figure 7 shows the plastic hinge distribution obtained for each rack structure model when subjected to incremental push load. Plastic hinges were observed in all models. However, a plastic hinge distribution over all the height of the structure was observed uniquely for SelCBS3. A concentration of plastic hinges in the up-rights of the first levels was obtained in the rest of models. Therefore, the pallet beams and up-rights are the main fuse elements in the rack structures studied.

In Table 6, the results of design and maximum shears are shown. Models with elastic behavior are subjected to shears greater than models with inelastic behavior. Furthermore, the design shears are notably lower than maximum shears for all cases. In Table 7, yield and maximum displacements, ductility, overstrength, ductility reduction factor, and response reduction factor in the cross-aisle and down-aisle directions are reported. The SelCBS3 model reached a R value of 3.99, similar to the value specified in [15]. However, other models reported values higher that R = 4 in both directions, mainly due to the overstrength contribution. The overstrength in the down-aisle direction is lower than in the cross-aisle direction, which is consistent with the capacity curves. Aditionally, the low values of

ductility reduction factors indicate that the R factor is controlled by overstrength, rather than ductility.

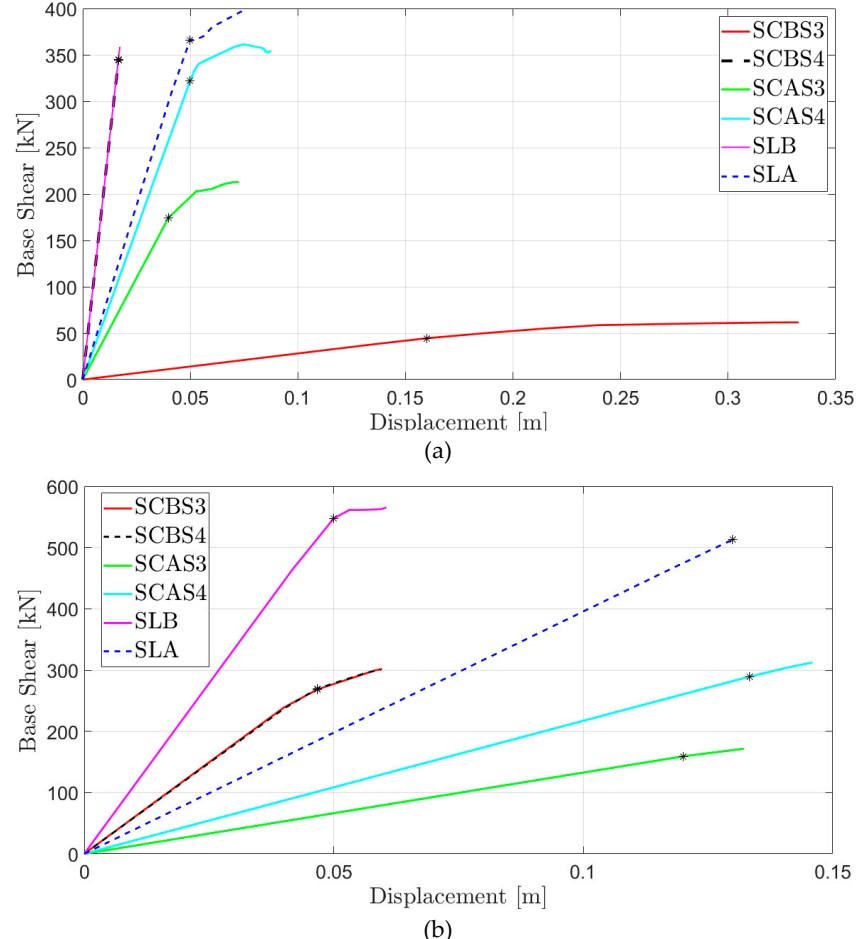

(a)

(b)

**Figure 6.** Capacity curves: (**a**) down-aisle direction and (**b**) cross-aisle direction.

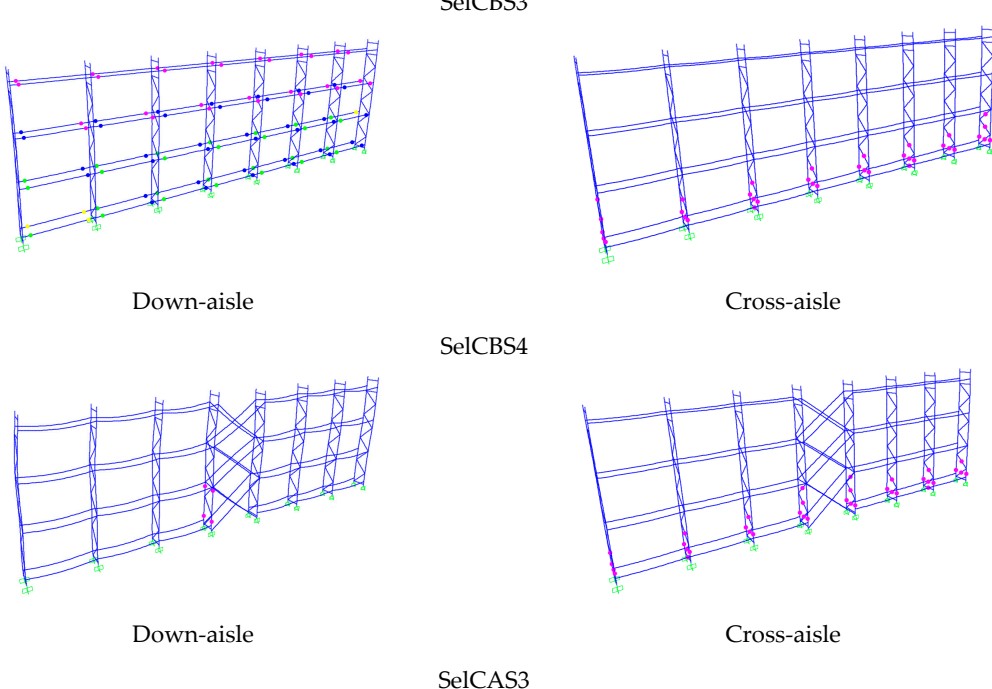

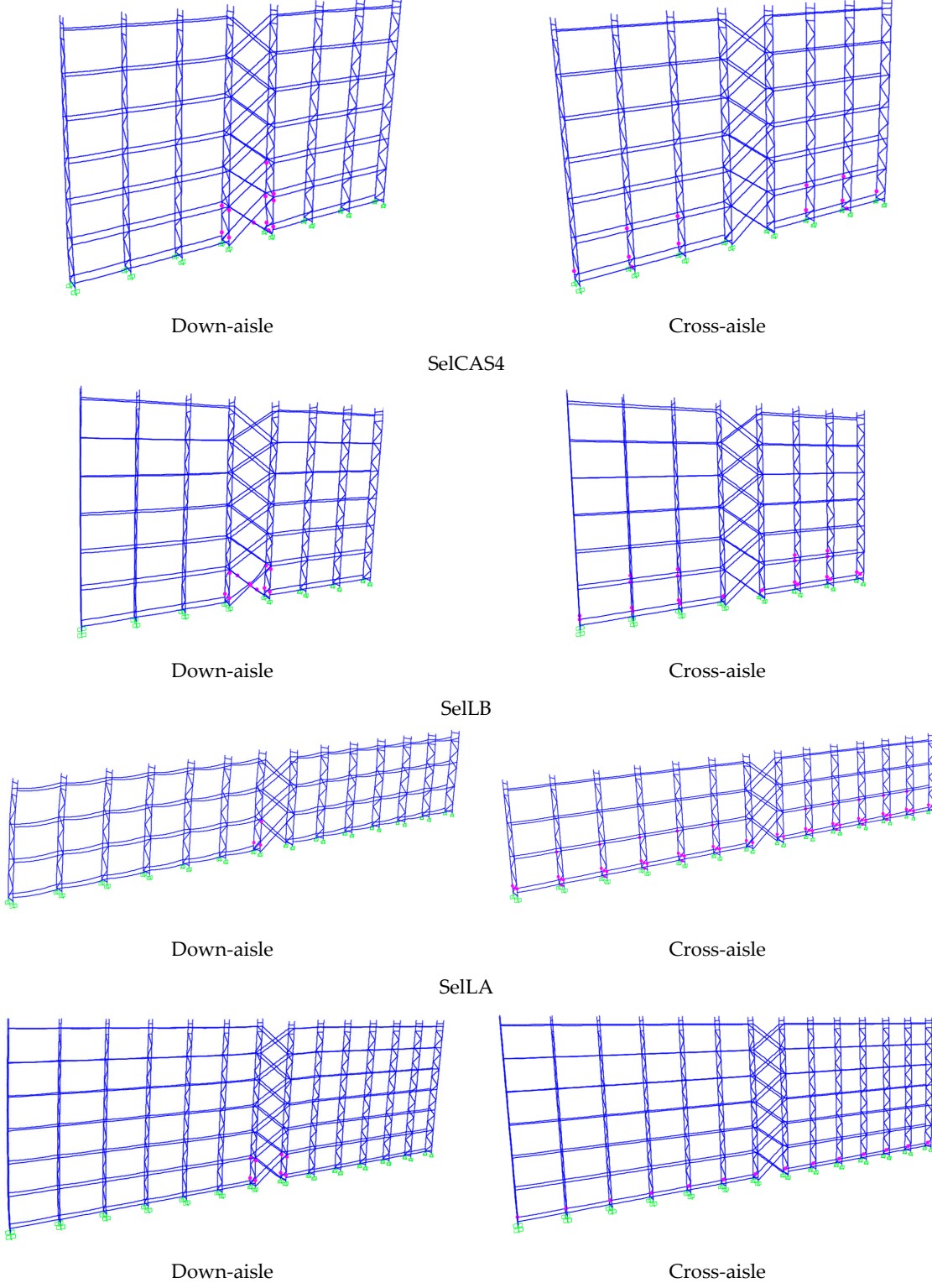

Down-aisle        Cross-aisle

SelCAS4

Down-aisle        Cross-aisle

SelLB

Down-aisle        Cross-aisle

SelLA

Down-aisle        Cross-aisle

**Figure 7.** Plastic hinge distribution in the down-aisle and the cross-aisle direction for pushover analysis.

**Table 6.** Base shear results.

| Model | $Vd_x$ [kN] | $Vd_y$ [kN] | $Vmax_x$ [kN] | $Vmax_y$ [kN] |
|---|---|---|---|---|
| SelCBS3 | 26.60 | 40.75 | 55.78 | 243.23 |
| SelCBS4 | 42.17 | 30.43 | 328.46 | 273.39 |
| SelCAS3 | 72.95 | 41.55 | 214.31 | 156.58 |
| SelCAS4 | 77.22 | 55.42 | 332.81 | 279.79 |
| SelLB | 84.34 | 68.68 | 354.97 | 502.65 |
| SelLA | 151.24 | 124.91 | 361.91 | 465.46 |

**Table 7.** Results of yielding displacements, maximum displacement, ductility factor, overstrength factor, ductility reduction factor, and response modification factor.

| Modelo | Δy_x | Δy_y | Δmax_x | Δmax_y | μ_x | μ_y | RΩ_x | RΩ_y | Rμ_x | Rμ_y | Rx | Ry |
|---|---|---|---|---|---|---|---|---|---|---|---|---|
| SelCBS3 | 14.35 | 2.45 | 33.13 | 4.68 | 2.31 | 1.91 | 2.10 | 5.97 | 1.90 | 1.68 | 3.99 | 10.04 |
| SelCBS4 | 1.11 | 3.14 | 1.79 | 6.06 | 1.60 | 1.93 | 6.75 | 7.79 | 1.49 | 1.69 | 10.06 | 13.17 |
| SelCAS3 | 2.61 | 7.56 | 9.90 | 13.42 | 3.79 | 1.77 | 4.50 | 5.78 | 2.56 | 1.60 | 11.52 | 9.24 |
| SelCAS4 | 2.84 | 8.12 | 7.99 | 14.35 | 2.82 | 1.77 | 6.79 | 7.97 | 2.15 | 1.59 | 14.61 | 12.66 |
| SelLB | 1.22 | 2.75 | 2.10 | 5.19 | 1.72 | 1.89 | 7.26 | 12.61 | 1.56 | 1.67 | 11.32 | 21.06 |
| SelLA | 2.64 | 6.65 | 7.49 | 13.04 | 2.84 | 1.96 | 7.51 | 11.69 | 2.16 | 1.71 | 16.22 | 20.01 |

*4.4. Nonlinear Synamic Response History*

A second approach to evaluate the seismic performance of rack structures was performed using nonlinear time history analyses. Each model was subjected to a set of seismic records. Nine ground motion records were selected with two horizontal components and magnitude M$_w$ between 7.7 and 8.8. The peak ground acceleration (PGA) with north-south (NS) and east-west (EW) components, time and magnitude are detailed in Table 8. The seismic records with EW component were applied in the down-aisle direction and NS component were applied in the cross-aisle direction. The number of records selected and methodology performed followed ASCE 41 [28], where a minimum of three records is required.

The response reduction factor R is obtained using the relation $R = R_\mu R_\Omega$, where $R_\mu$ = Ve/Vmax, is the ductility reduction factor (elastic shear divided by maximum shear), and $R_\Omega$ = Vmax/Vd, is the overstrength of the structure (maximum shear divided by design shear) obtained from pushover analysis. For all cases, the story drift is calculated and reported as a measure of displacement control. Figure 8 shows the pseudo acceleration response spectra of all records considered, which were obtained using Nigam and Jennings method [29], with a 5% damping ratio. The design spectra for soil types 3 and 4, according to [15], are also included in this figure.

**Table 8.** Chilean ground motion records.

| Epicenter | Date | Mw | Station | N° | Name | $\Delta t$ [s] | Duration [s] | PGA [g] |
|---|---|---|---|---|---|---|---|---|
| Taparacá | 13-6-2005 | 7.8 | Pica | 1 | picaEW2005 | 0.005 | 252 | 0.735 |
| | | | | 2 | picaNS2005 | | | 0.544 |
| | | | Iquique | 3 | iquiqueEW2005 | 0.005 | 196 | 0.227 |
| | | | | 4 | iquiqueNS2005 | | | 0.217 |
| Tocopilla | 14-11-2007 | 7.7 | Mejillones | 5 | mejillonesEW | 0.005 | 218 | 0.141 |
| | | | | 6 | mejillonesNS | | | 0.42 |
| Cobquecura | 27-2-2010 | 8.8 | La Florida | 7 | lafloridaEW | 0.005 | 208 | 0.133 |
| | | | | 8 | lafloridaNS | | | 0.186 |
| | | | Puente Alto | 9 | puentealtoEW | 0.01 | 147 | 0.268 |
| | | | | 10 | puentealtoNS | | | 0.266 |
| | | | Hospital Curicó | 11 | curicoEW | 0.01 | 180 | 0.414 |
| | | | | 12 | curicoNS | | | 0.475 |
| Iquique | 1-4-2014 | 8.2 | Iquique | 13 | iquiqueEW2014 | 0.005 | 297 | 0.316 |
| | | | | 14 | iquiqueNS2014 | | | 0.202 |
| | | | Pica | 15 | pica2014EW | 0.005 | 286 | 0.335 |
| | | | | 16 | pica2014NS | | | 0.279 |
| Illapel | 16-9-2014 | 8.4 | Monte Patria | 17 | MontePatria2015EW | 0.005 | 470 | 0.831 |
| | | | | 18 | MontePatria2015NS | | | 0.713 |

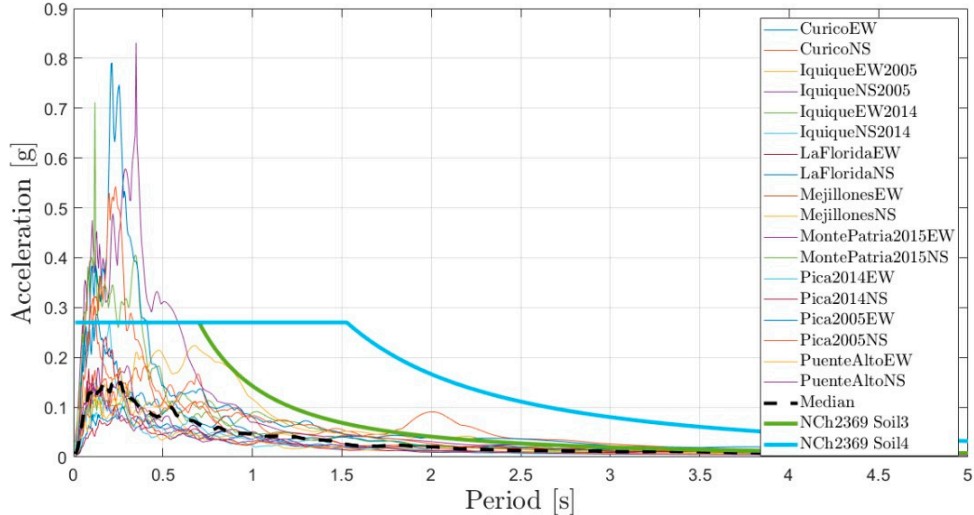

**Figure 8.** Pseudo acceleration response spectrum curves.

*4.5. Results of Nonlinear Dynamic Response History Analyses*

Figures 9–11 show the plastic hinge distribution obtained for each rack structure model when subjected to the different seismic records. Plastic hinges were observed only for models subjected to Curicó, MontePatria, Mejillones, and Pica 2005 seismic records, while an elastic behavior was observed for other seismic records. Therefore, only the results for the former records are presented in these figures. In particular, a plastic hinges distribution overall the height of the structure was observed for SelCBS3-MontePatria, SelCBS3-Curicó, and SelCBS3-Mejillones. A concentration of plastic hinges in the first level was observed in the rest of the models. Only for the SelCAS4-MontePatria, SelLB-MontePatria, and SelLB-Pica2005 models were the plastic hinges concentrated in

the braces of the bracing tower. Therefore, the pallet beams and up-rights are the main fuse elements in the rack structures studied.

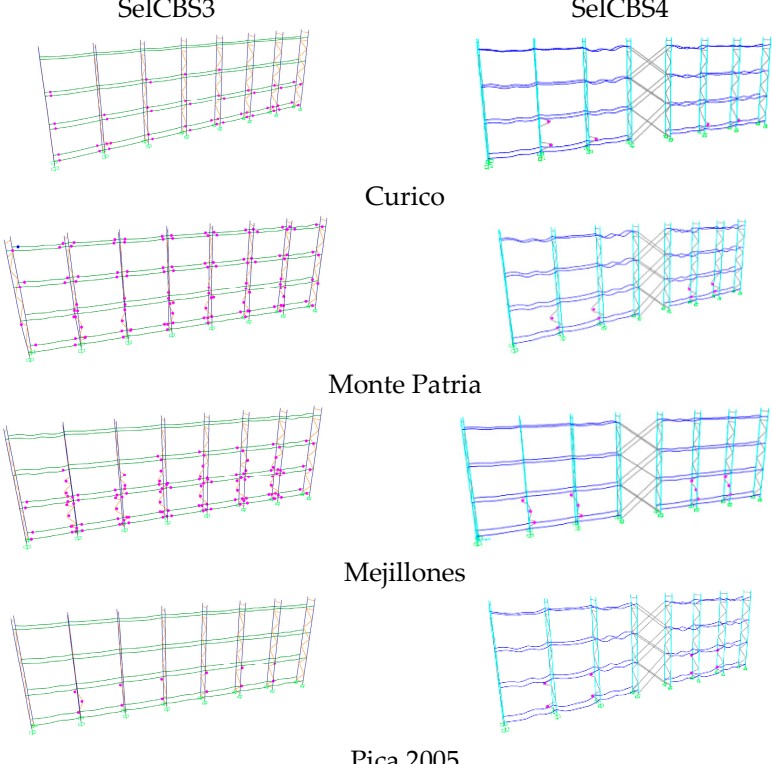

SelCBS3      SelCBS4

Curico

Monte Patria

Mejillones

Pica 2005

**Figure 9.** Plastic hinge distribution in down-aisle and cross-aisle direction for SelCB prototypes.

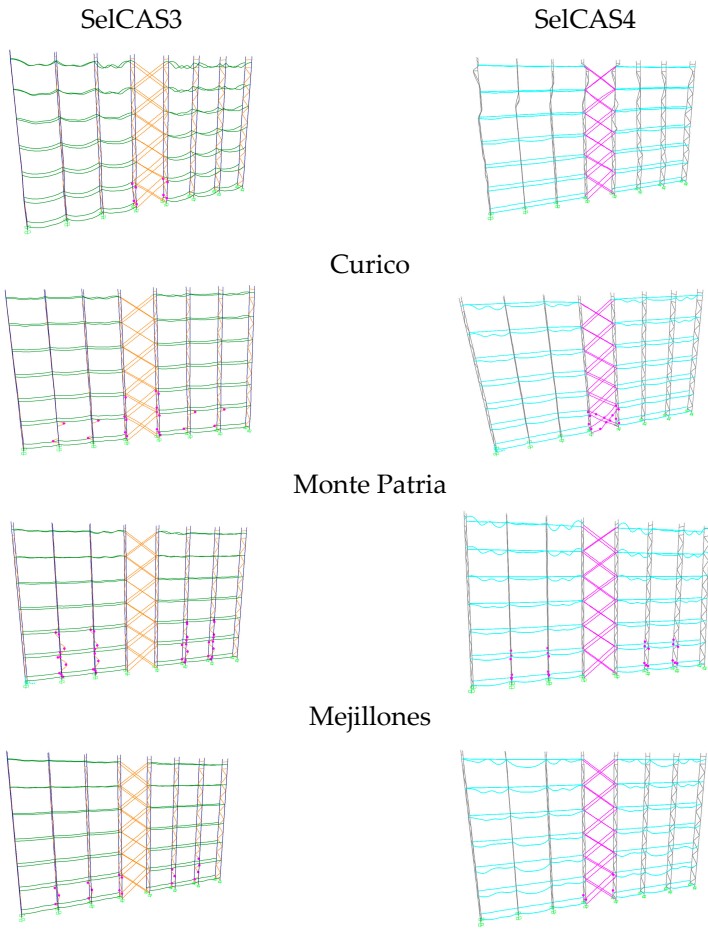

SelCAS3      SelCAS4

Curico

Monte Patria

Mejillones

Pica 2005

**Figure 10.** Plastic hinge distribution in down-aisle and cross-aisle direction for SelCA prototypes.

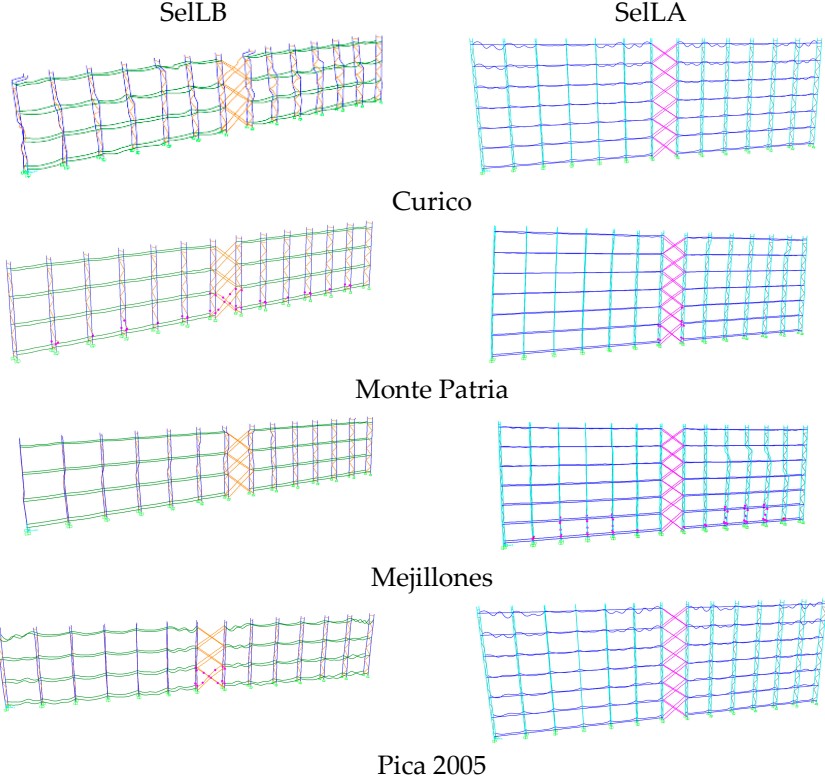

Pica 2005

**Figure 11.** Plastic hinge distribution in down-aisle and cross-aisle direction for SelLB and SelLA prototypes.

In Figure 12, the maximum story displacements in down-aisle and cross-aisle directions are reported. In the down-aisle direction, maximum displacements of up to 0.20 m were reached in the SelCBS3 model. The rest of the models reached displacements in the range of 0.05 m to 0.10 m, due to the influence of the bracing towers, which provide a great lateral rigidity to rack structures. In the cross-aisle direction, maximum displacements of 0.15 m were reached in all models. In addition, a significant reduction of 50% in displacements was obtained in the SelCBS4 and SelLB models, in comparison to the SelCAS4 and SelLA models, due to the influence of global slenderness in the inelastic response.

In Figure 13, the maximum story drift ratios are reported in the down-aisle and cross-aisle directions. In the down-aisle direction, a maximum drift ratio of 3.3% is reached in the SelCBS3 model, although was designed for 1.5%, according to [15]. However, a maximum drift ratio of 0.27% was obtained in the SelCBS4 model, due to the incidence of great rigidity provide by the bracing towers. The rest of models reached drift ratios of less than 1.5%, as established in [15], except for the SelLA model, which reported a slightly higher value. In the cross-aisle direction, drift ratios lower than the limit imposed by [15] were obtained in the SelCBS3 model, except the results obtained for the Mejillones seismic record, which reached 1.55%. A similar response was achieved in the rest of models. Additionally, the models with low slenderness, such as SelCB and SelLB, reached drift ratios lower than 1% in comparison to the SelCA and SelLA models, showing the influence of global slenderness in the inelastic response.

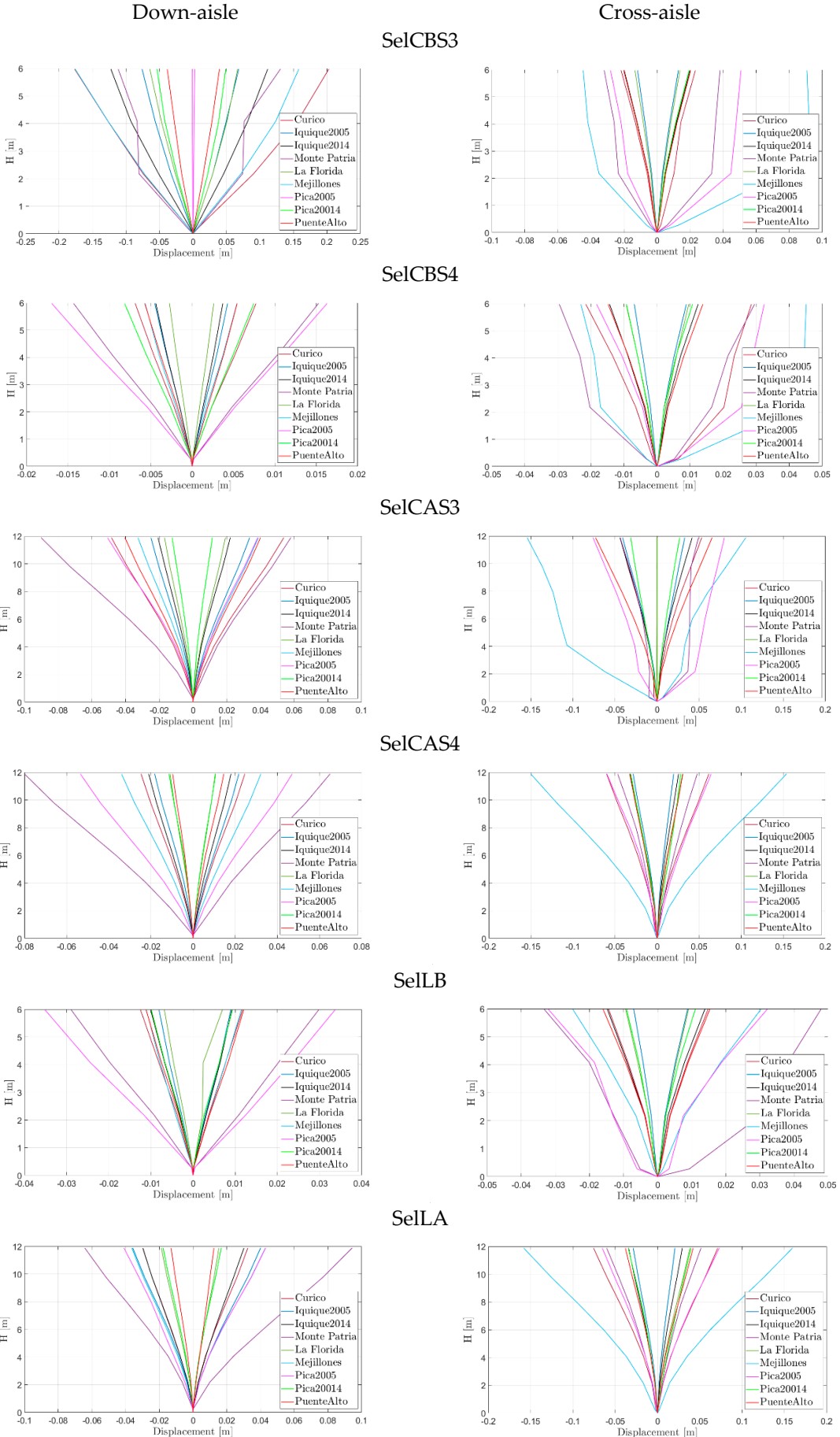

**Figure 12.** Maximum story displacements for nonlinear time history analysis.

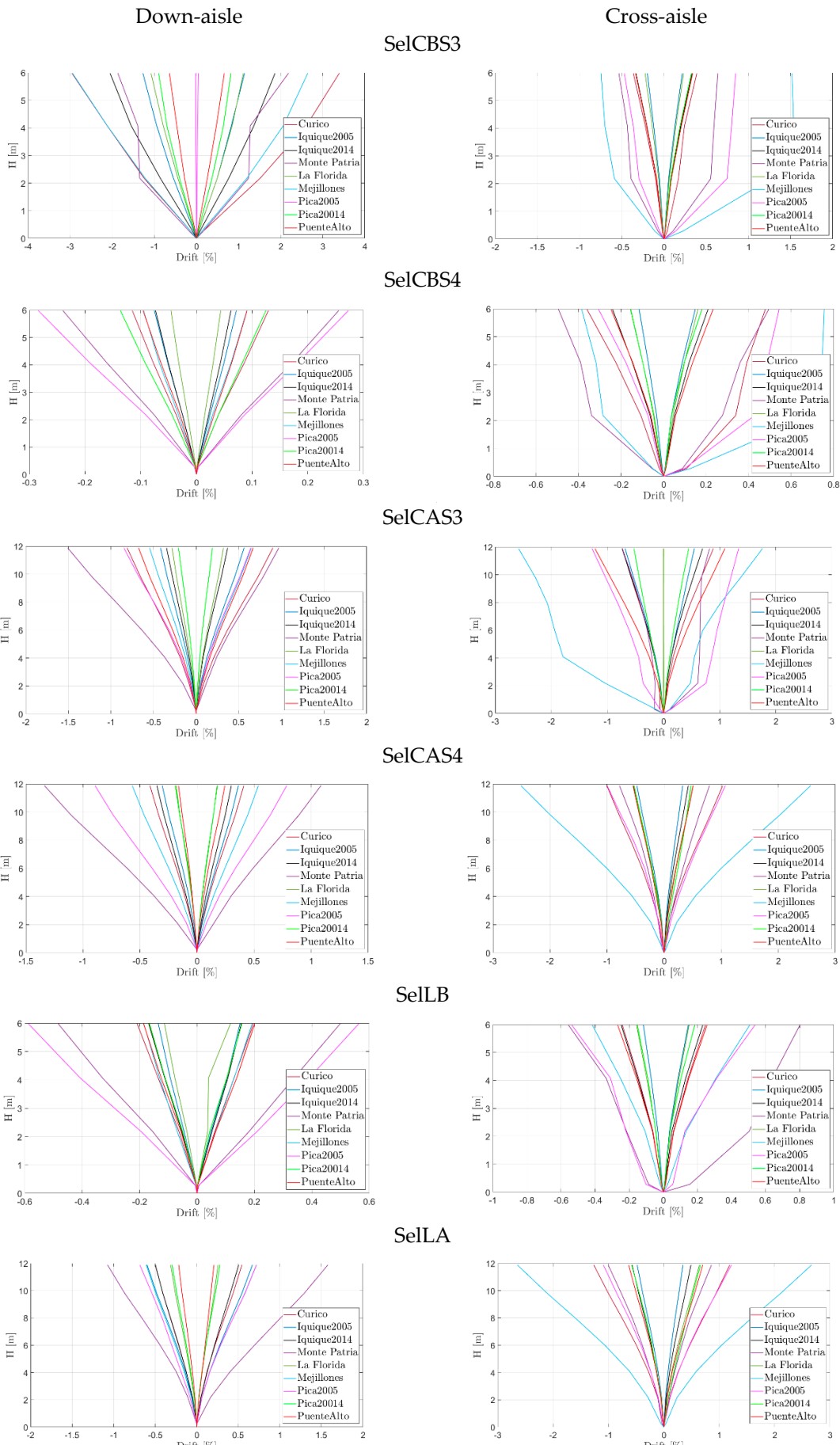

**Figure 13.** Story drift ratio for nonlinear time history analysis.

In Figure 14, a seismic coefficient ratio C = Vmax/W is reported, where Vmax is the maximum shear obtained for each seismic record divided by the weight of rack structure. This parameter was calculated for the down-aisle and cross-aisle directions in rack models (1: SelCBS3, 2: SelCBS4, 3: SelCAS3, 4: SelCAS4, 5: SelLB and 6: SelLA). The Cmin and Cmax values are the minimum and maximum seismic coefficients, respectively, obtained according to [15]. In the down-aisle direction, the C values are higher than the Cmax values indicated in [15], except for the values obtained in the SelCBS3 model, where a good adjust is obtained. In cross-aisle direction, a great dispersion was obtained. However, several values are higher than the Cmax imposed by [15]. Finally, the Curicó, MontePatria, Mejillones, and Pica2005 seismic records reached the majors PGA of set records, which is proportional to the Vmax obtained in models subjected to these seismic records.

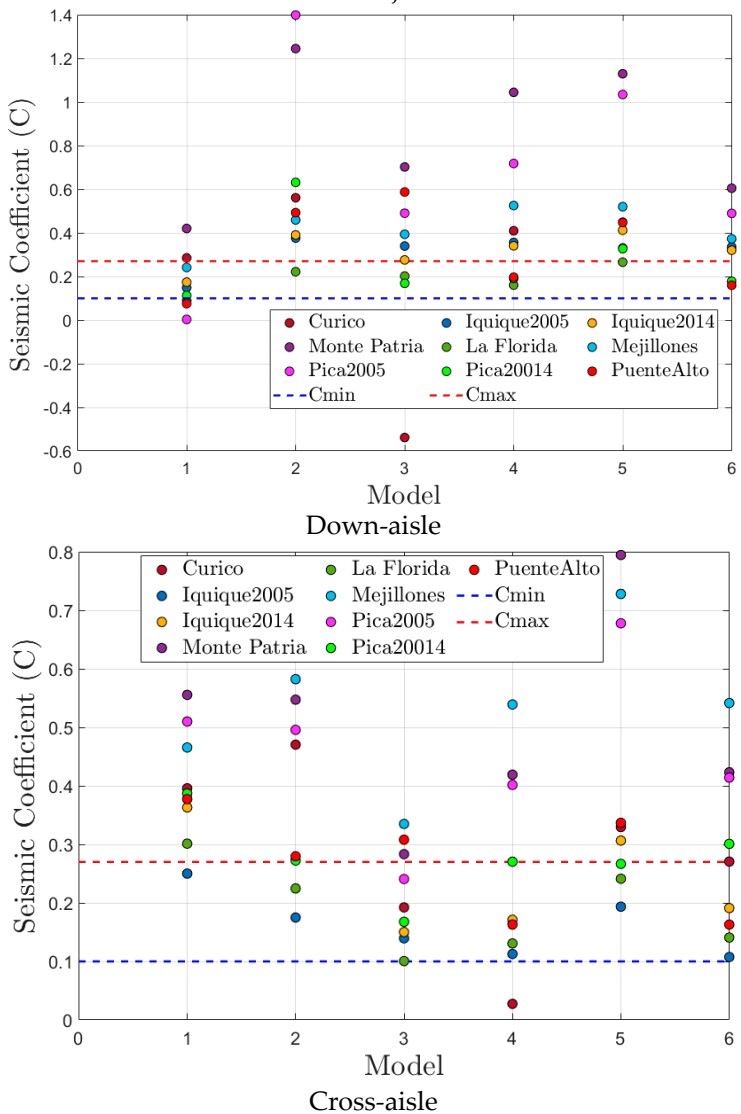

**Figure 14.** Summary of maximum seismic coefficient for nonlinear time history analysis.

In Figure 15, a response reduction factor R is reported en both directions. A reduction factor R = 4 is established in [15] for rack structures. In the down-aisle direction, a range of 2 < R < 4 was obtained for the SelCBS3, SelCAS3, and SelLA models. In adition, a range of 4 < R < 8 was obtained for the SelCBS4, SelCAS4, and SelLB models. In the cross-aisle direction, the reduction factor R reported reached higher values than R = 4 imposed by [15]; however, a great dispersion is obtained. Finally, the R values obtained are controlled by overstrength more than ductility.

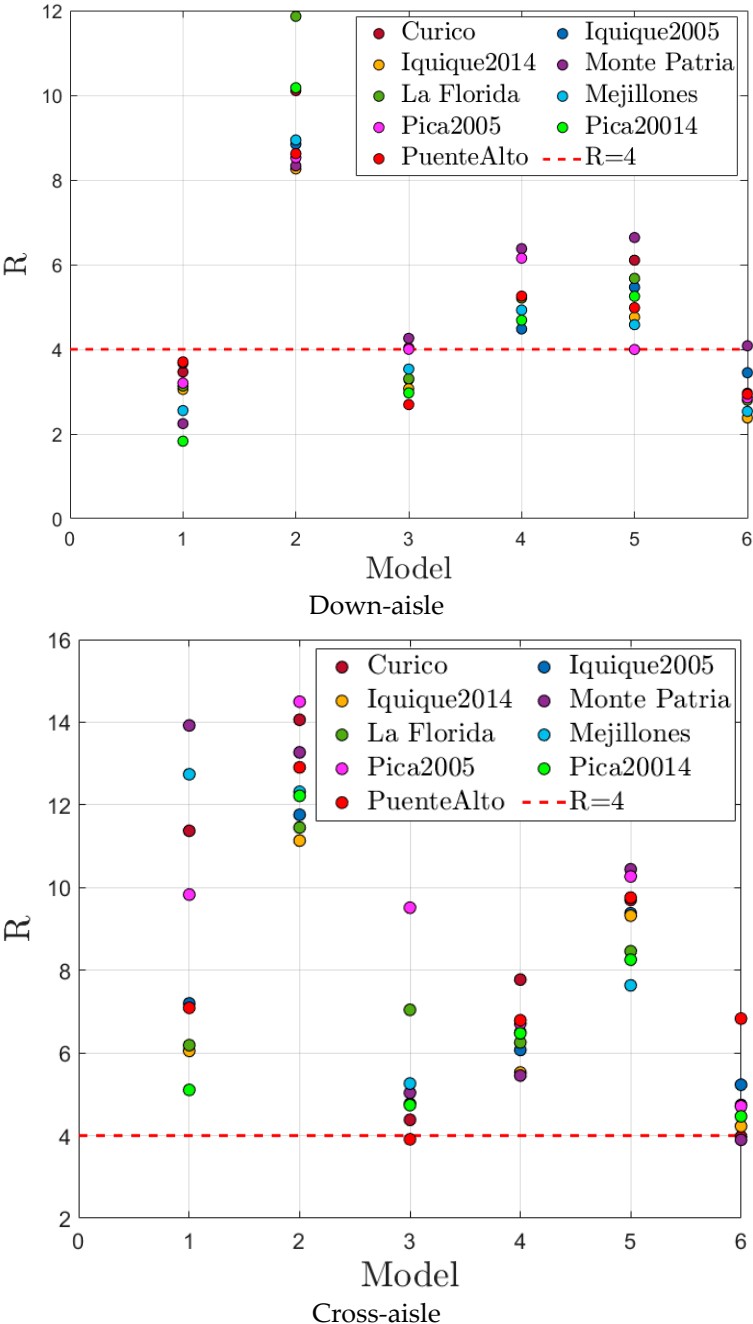

**Figure 15.** Summary of response reduction factor R for nonlinear time history analysis.

## 5. Conclusions

In this paper, the seismic behavior of the selective steel storage racks with variable slenderness using the nonlinear pushover and nonlinear dynamic time history analysis was studied. The connection behaviors in the numerical model were considered. From selected Chilean ground motion records, a total of 54 models were studied. Additionally, a previous seismic design using modal response spectrum analysis was reported, showing the influence of conditions imposed by Chilean Standard of Industrial Structures Seismic Design in the seismic performance. It also revealed the impact of the bracing tower on the inelastic behavior of rack structures. The results showed an elastic behavior, mainly in the cross-aisle direction, in comparison to the down-aisle direction. The inelastic action was concentrated in the pallet beams and up-rigths; however, plastic hinges occurred in braced tower braces of the SelCAS4 model when the MontePatria seismic record was applied. Higher values

of base shear were reached, due to an elevated rigidity in rack configurations. However, an acceptable behavior was obtained. The response modification factor R calculated is influenced by the overstrength obtained from seismic design. In the pushover analysis, the response reduction factor R values obtained in the cross-aisle and down-aisle directions are higher than the R = 4 specified in Chilean code, except in the SelCBS3 model, which reached a R = 3.99 in the down-ailse direction. However, the R values obtained in the nonlinear dynamic analysis are below of R = 4 in SelCBS3, SelCAS3, and SelLA models exclusively in the down-aisle direction. Therefore, future researchs should be concentrated in down-aisle direction, with large aspect ratios and new limits on the use of selective steel storage racks being considered.

**Author Contributions:** Conceptualization, E.N.; methodology, E.N.; software, C.A. and E.N.; validation, E.N.; formal analysis, C.A.; investigation, E.N., R.H. and C.A.; writing—original draft preparation, E.N.; writing—review and editing, R.H.; visualization, R.H.; supervision, E.N. All authors have read and agreed to the published version of the manuscript.

**Funding:** This research received no external funding.

**Acknowledgments:** The authors would like to thank Lemusse Company, Tomé, Chile for its technical support.

**Conflicts of Interest:** The authors declare no conflict of interest.

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
