# Peer review of "Assessment of the Seismic Behavior of Selective Storage Racks Subjected to Chilean Earthquakes"

_metals, doi:10.3390/met10070855_

Round 1

Reviewer 1 Report

The paper "Assessment of the seismic behavior of selective storage racks subjected to Chilean Earthquakes" concerns issue of designing structures in the light of safety and beahaviour under nature-based model of seismic conditions. Thus the suject of the work is both vital and important. The work presents also an interesting optimization problem as for the effect of structure consideration on displacements and performance of the steel strage racks. However, besides the meaning of the material aspect in the plan of the study and correctness of modelling results, the problem and the results boil down to structural mechanical problem with little technology/material science aspects.  In the favour of publication of this research in metals journal is the aspect of considering cold-formed profiles, where geometry and material considerations are properly taken into account. Thus the quaestion arises for cognitive aspect of the case study presented in the work. It should be enhanced.

Beyond this there are issues which need to be corrected for improvement of the article quality: 

1) Because of numerous variants of seismic action and the configurations of the analysed racks, a graphs or table summarizing the range of analysis is advised

2) it is advised that the results in fig. 8. are given in a from which allows for direct comparison, e.g. different types of rack configuration next to one another for the same seismic mode (suggestions are shown in attached scans)

3) Figs 9 and 10 can fit to page (not splitted between pages); a table-like layout will help with headings (titles, e.g. SelCBS3 etc.) moved on the left to the diagram. The legends are too small to be readible - moving enlarged legend next to the diagrams on the rigth (one will do as it's the same for down-aisle and cross-aisle).

4)  The evolution of the modeling methods is well described in the introduction which is a good background for selection of the models used in the work. However, the meaning of non-linear approach should be addressed with bigger emmphasis on its meaning on the credibility of the results. It should also be mentioned in conlusions.

5) The paper needs  language corrections, in the range of

  • style:  lines 26,33,39,47-50,175, 205,223, 281,303, 326,342/344, 258,
  • spelling/grammar:52,168,285,294,306,
  • typos and other:140,141,0142,tab. 2., 165,214,229, tab.5., 242,249, 259, tab.7 (modelo), 301,377, 397, 436.
  • unification of definition of parameter R

A scan of hadwritten notes is attached below, (despite the fact it was not meant for sharing, so please excuse the quality) in case it turns out helpful.

Author Response

Metals, Special Issue "Advances in Structural Steel Research"
MDPI

Subject: Revised Manuscript “metals-836580”

Dear Editorial Staff:

Please find enclosed the updated version of the paper entitled Assessment of the seismic behavior of selective storage racks subjected to Chilean Earthquakes by Eduardo Nuñez, Catalina Aguayo and Ricardo Herrera. This draft has been extensively revised to address the comments received following the initial review of the paper. The authors would like to thank the reviewers for their comments, which were very helpful in improving the quality and clarity of the manuscript. Attached is a summary of how the recommendations of the reviewers were addressed in the revised version of the paper.

Sincerely,

Reviewer 2 Report

English language needs improvement. At least, a spell check should be performed before handing in the paper!!! Examples:

Line 125: clasiffied

Line 171: degress

Line 185: sumary

and many more.

The introduction lists multiple references. It also gives a short explanation of the method used in the paper. It does not explain though, why this paper is necessary and what  limitation lies in the existing research (line 122). I suggest to add more distinct information on the research goal. Why is the chosen method relevant?

The most relevant elements for plastic response of rack structures are the connections. They are not shown at all. Neither are their failure modes. The structural properties are taken from literature. A closer view would be helpful to assess the used properties. Explain and prove: Do the models used in calculation represent the chosen rack system?

Figure 6 and others: I suggest to use black lines, dottet, dashed or others, in graphs. Especially, yellow lines on white are extremely hard to work with. Graphs are generally too small - fonts in graphs are not readable - must be increased.

Numbering of figures is wrong: figures in lines 231, 267 and 269 are all called figure 6.

Think about reducing the number of graphs explaining the results. Having e.g. figure 8 spanning two pages is not acceptable. Especially, since the single graphs are not explaind each on its own. It is a lot better to just show few representative graphs and explain them well.

The conclusions only contain a summary of the calculation results, no evaluation, no suggestions for implementation of the results in practical or scientific use. This would be a very interesting addition.

The explanation of the model numbering from 1 to 6 must be given before figure 11, not after!

Author Response

(The authors gave the same response as above.)

Reviewer 3 Report

This paper contains scientific information very useful for engineering design against erthquakes. I should be published. However, before being accepted for publication, many figures should be improved (in appearance) to reach mdpi standards, in particular enlarging their labels.

Author Response

(The authors gave the same response as above.)
